# New Perspective of Cultural Sustainability: Exploring Tourism Commodification and Cultural Layers

Lingxiao Bai [1] and Shixiu Weng [1,2,*]

1   School of Geography and Planning, Sun Yat-sen University, Guangzhou 510006, China;
    bailx@mail2.sysu.edu.cn
2   Center for Tourism Planning and Research, Sun Yat-sen University, Guangzhou 510275, China
*   Correspondence: wengshx8@mail.sysu.edu.cn

**Abstract:** Commodification is an important research issue regarding cultural sustainability. This paper draws on the theory of cultural layers to understand tourism commodification and the development of local cultures. Through 76 days of field investigation and about 60 face-to-face interviews, this paper explores the characteristics of tourism commodification in festivals and rites related to sacrifices to Genghis Khan and the relationship between its five layers of commodification. Research findings reveal that commodification mainly occurs in the layers of artifacts and patterns of behavior. In addition, the process of tourism commodification does not necessarily follow the sequence of cultural division from the outside to the inside; instead, the deeper layer of commodification will inevitably drive the commodification of its outer circle, and different layers of commodification display different social problems as well. It is also found that this theory is suitable for other tourism commodification cases examined in existing studies. Taken together, analyzing tourism commodification through layered thinking can unveil the degree of commodification, offer an avenue to address the extant debate over the pros and cons of tourism commodification, and provide a basis for guiding the regulation of tourism commodification more effectively.

**Keywords:** tourism commodification; layers of culture; Genghis Khan sacrifice

## 1. Introduction

Culture is an important part of the product offered to tourists. With the growing prosperity of the tourism industry, ethnic tourism has been a focus of many nations and states in their promotion of economic and cultural development [1–3] and has become an important means of local economic development in ethnic areas [4]. Meanwhile, the issue of cultural commodification has also become increasingly prominent with the significant rise in tourism [5]. Thus, commodification is an important research issue in cultural sustainability.

Culture enables development through cultural heritage, industry, infrastructure, and tourism. Cultural commodification can either contribute to cultural sustainability or hinder it. With regard to the phenomenon and problems of cultural commodification in tourism development, relevant research mainly focuses on the impact of cultural commodification and the relationship between cultural commodification and authenticity. Research on the impact of cultural commodification covers both the positive and negative impacts. Studies advocating positive impacts hold that cultural commodification driven by tourism development can preserve traditional culture [6], promote cultural revival [7], enhance ethnic identity [8], and increase ethnic pride [9]. It provides a way for the publication or raising of cultural values [10,11] and can also improve the community members' ability to increase incomes [12,13] and improve the well-being of the community [14,15]. There are also studies suggesting that in some countries, marginalized cultures have adopted tourism as a political tool to construct their identity [16], thus providing local residents with controllable political resources [17]. Researchers arguing negative impacts focus on exploring cultural

performances and variations in folk functions in the context of tourism [18], as well as the conflict between ethnic and cultural traditions and commodification [19].

Relevant studies reveal that indigenous tourism for economic benefits, namely, cultural commodification, has brought about such risks as the erosion of rights, being subject to suppression and exploitation [20,21], and weakening the intrinsic value of culture [22,23] for local communities, leading to the loss of the significance of culture for local residents [24], the loss of the sanctity of traditional culture [25,26], and even the displacement and marginalization of indigenous peoples [27]. The commodification process may also subvert the locals' control over cultural expressions [28] and cause tension and conflicts between local residents and tourism operators [29,30]. The relationship between cultural commodification and authenticity is one of the most talked-about and enduring topics in the research on tourism [24,30–43]. Since MacCannell initiated a debate on the relationship between authenticity and tourism, authenticity has always been at the core of discussions about social and cultural consequences. As argued by him, the performances focusing on folk cultures presented to tourists are simply "stage authenticity" organized by culture suppliers, who change the nature of the product for the sake of creating an attractive packaging surface [24]. Greenwood criticized that cultural commodification has changed the meaning of cultural products and practices to the extent that cultural products ultimately become meaningless to producers, which is a disregard and desecration of cultural authenticity [44]. However, some scholars view the topic from a dialectic perspective and hold that authenticity is not an absolute concept but a relative, interpretive, and socially constructed concept [6,45,46]. According to Wang, the authenticity of tourists' experience has three sources, namely, objectivism, constructivism, and postmodernism [44]. The perception of authenticity is a dynamic, flowing, negotiated, and creative process, and evaluation changes along with the specific context and personal perspectives. Adams proposed that culture and authenticity are manifestations of tourists and hosts attempting to reflect their own desires in their interaction process [47]. Therefore, the authenticity of the tourism environment is not a real estate or tangible asset but, instead, an observer's judgment and evaluation of the environment [48]. Some researchers suggest that if authenticity is equivalent to originality, real authenticity will then only exist in cultural symbolization for the first time [49]. Therefore, they call for abandoning this concept and term since they have no common ground in existence, meaning, or significance [39].

However, previous studies have either analyzed culture as a whole by discussing overall changes and the impact of tourism on the overall picture [9] or only analyzed one element to represent the overall picture [50]. In fact, culture is a complex organism composed of many elements; thus, both the whole analysis and one-element analysis cannot fully reveal the degree of cultural commodification and end up losing the significance of the horizontal comparison between multiple cases. In this sense, a deep exploration of a typical case to reveal its internal structure and characteristics is worthy of study.

Due to the different development levels of tourism, the degree of cultural commodification varies in different regions. Cultural commodification is a form of cultural change and an important topic in the study of cultural sustainability. Culture is considered to be the fourth pillar of sustainable development and has the same importance as the three known pillars (social, economic, and environment) of sustainable development [51]. However, the concept of "culture" encompasses a variety of elements, which can be either explicit or implicit. The analysis of the internal structure of cultural commodification is, in essence, the reveal of the internal levels of culture. Only by clarifying the layered structure of cultural components can we understand cultural commodification in a systematic and comprehensive way. In order to have a more intuitive understanding of the layers of cultural commodification, this paper, by taking the religious sacrifices to Genghis Khan as a case study, revealed the layered structure of cultural commodification by analyzing its commodification at various layers. The sacrifices to Genghis Khan are an ancient traditional sacrificial rite of Mongols with a history going back nearly 800 years. The sacrifices to Genghis Khan are rich and mysterious in contents and represent the highest standard of

sacrificial rite of Mongols and the supreme belief of Mongols, having a long-lasting and profound impact on Mongols. Given its clear and comprehensive structural levels and its long history, this sacrificial culture is a typical case for both diachronic research and research on cultural commodification. Therefore, by selecting the sacrificial ceremony for Genghis Khan held by Mongols as the study case, this paper explores the commodification level of the sacrifices to Genghis Khan by adopting the cultural layers theory, with an emphasis on tourism commodification, and attempts to expand the scope of application and significance of this theory.

## 2. Theoretical Review

In the eyes of anthropologists [52,53] and organizational researchers [54], culture is a set of shared cognitions among members of a social unit. The concepts of shared values, shared understanding, patterns of beliefs, and expectations are the foundation of researchers' views on the nature of culture. Organizational researchers not only conduct similar conceptual analyses of culture but also evaluate different elements. Early-stage researchers simply divided culture into two layers, namely, the lower layer of culture and the upper layer of culture (cultural consciousness). In later research, scholars gradually discovered that cultural layers are like a system, with diverse classification methods available and the ability to further subdivide under existing classifications. The purpose of various classification methods is basically the same; namely, they attempt to figure out the organizational relationships and operational logic between various elements of culture. Malinowski, a cultural anthropologist of functionalism, put forward the theory of the "tripartite structure of culture", which divides culture in a broad sense into three levels: the material level, institutional level, and spiritual level [55]. Pang divided the cultural layers into material, relational, and ideological levels [56]. However, the subjectivity or objectivity of these elements varies, and the observability and availability are also diversified as to different researchers and organizational members. Therefore, the inconsistent and chaotic evaluation of cultural elements has always troubled many organizational culture researchers [57]. In addition, in previous studies, the term "culture" often had different connotative meanings, such as being used to define a specific social unit and to describe specific social processes [58]. This is not attributed to the significant differences in the definition of culture among researchers but rather the different types of data collected by researchers. Some researchers focus on unconscious assumptions implicit in the words and actions of organizational members [59], some focus on observable values in events or rituals [60], while others focus on behavioral norms attached to social units [61], and many descriptive articles talk about material artifacts of organizational life [62,63]. These studies are all aimed at examining the potential of cultural products; namely, they conform to the definition of culture in organizational research to some extent, but their focus is radically different.

In this regard, in order to put the focus of cultural research and operationalization on relevant factors, Rousseau proposed viewpoints on the stratification of culture [64]. Upon examining the main elements of culture, he divided them into five layers in a circular form from easily accessible to difficult to assess, namely: artifacts, patterns of behavior, behavioral norms, values, and fundamental assumptions. When the research objective is to investigate the material artifacts and other material manifestations of social systems or the activities and patterns of interaction among organizational members, the levels of artifacts and patterns of behavior become the main elements of culture. On the contrary, if we choose to focus on deeper levels of culture, the cultural levels related to values, beliefs, and expectations will become the main content of cultural concepts. Individuals construct the meaning of events through the recognition of patterns of behavior, thereby developing an organized worldview, which is a constructed worldview; behavioral norms are an example of the social construction experienced by the members of a unit, requiring reciprocity and sharing as the basis for their existence; values are preferences that are typically manifested in observable behaviors; the fundamental assumption is the deepest

and most subjective element in culture, which may affect all other elements [65]. Values and fundamental assumptions are the cultural levels that are most difficult to reveal. A fundamental assumption refers to "unconscious, taken for granted beliefs, perceptions, thoughts, and sensations (the ultimate source of value and action)" [66], which illustrates the complexity of this concept and the difficulties in understanding it subsequently.

Rousseau's theory model provides a reference for current research on social impact. Deery et al. adapted Rousseau's concept of cultural layers to promote research on the social impact of tourism [67]. They indicated that this framework could test the perception of social impact on artifacts, behaviors, values, and fundamental assumptions. Currently, the focuses of recent research on the social impact of tourism are the artifacts. Traffic congestion, disruptions to normal lifestyles, increased entertainment opportunities, and protection or destruction of heritage are all factors related to the positive and negative impact of the tourism industry on communities. The research on the social impact of the tourism industry until now has effectively addressed the impact at this level, especially regional innovation and investment image assessment under conditions of sustainable transformation, which is a logical component of regional sustainable development policies [68]. The understanding of the impact of tourism on society at the next level requires further studies. Understanding the patterns of behavior of residents and tourists helps with understanding why certain impacts are more important to residents when compared with other impacts. By exploring the behavioral norms of residents and tourists based on the understanding of patterns of behavior, it was found that both parties have a common understanding or lack of understanding of what constitutes acceptable behaviors [69]. This is proven by the changing behaviors of residents living in Beijing's Hutongs, along with time zones and the cycles of the four seasons [70].

The cultural layers theory is an abstract expression of the internal relationship of culture as an organism, revealing the operating principles and internal contradictions of culture in the intrinsic logic of culture. Therefore, cultural layers can be inferred as the essence of culture. In fact, the sense of cultural stratification was developed by scholars in the early years, and different scholars presented different layers.

The commodification of tourism, as a form of cultural change, has been shaped by multiple stakeholders and comprehensively reflects internal and external factors, making it an extremely complex issue. The cultural layers concept of organizational culture research may be a good perspective for tourism researchers to analyze the cultural impact of tourism, which can help us more clearly judge the degree of the commodification of cases utilized by tourism. Compared to stratification into two and three levels, the author believes that Rousseau's five-level stratification method can more comprehensively and systematically reveal the complex structure and contradictions of tourism commodification. Therefore, this article creatively introduces Rousseau's cultural layers theory into the study of tourism commodification [64]. Based on the five layers, the commodification level of the sacrificial ceremony of Genghis Khan was analyzed, and the necessity and applicability of conducting a layered analysis of tourism commodification were explored.

## 3. Case and Research Methods

### 3.1. Case Introduction

The sacrificial ceremony of Genghis Khan came into being after his death in 1227. His descendants placed portraits and relics symbolizing the soul of Genghis Khan in a white royal tent, and dedicated guardians were arranged to guard and worship Genghis Khan throughout the country. These guardians were called "Darkhad", meaning "men with a sacred mission". Later, with the successive deaths of Genghis Khan's family members, the quantity of sacrificial white tents and relics gradually increased, forming the "Eight White Houses", namely, eight white royal tents (Mongolian yurts), with the Great Khan's White House at the core. The Eight White Houses of Genghis Khan form the spiritual sanctum of Genghis Khan, the symbol of Genghis Khan. It is believed that the Eight White Houses are the place where Genghis Khan's soul returns and the consecrated place that the Mongol

khans of all previous dynasties must worship. This special cultural form displays the traditional customs, rituals, beliefs, concepts, language, writing, cultural arts, and other aspects of the Mongolian ethnic group. The sacrificial ceremony of Genghis Khan was based on the primitive Shamanistic culture. In terms of sacrificial contents, it mainly expresses the worship of Mongke Tangri, their ancestors, and heroic figures; in terms of sacrificial forms, it mainly reproduces the ancient Mongolian forms of fire sacrifice, milk sacrifice, wine sacrifice, animal sacrifice, and song sacrifice; in terms of sacrificial utensils, it mainly uses a variety of precious utensils with distinctive features as well as many sacrificial texts and poems, blessings, and sacrificial songs to demonstrate the artistic and aesthetic attributes of grassland ethnic groups towards the overall nature of the region and animals. These sacrificial poems take a poetic style and are passed down from generation to generation in the oral literary form, possessing a high literary value.

The sacrificial ceremony of Genghis Khan is the highest level of worship of the Mongols and still retains all of the aspects of the sacrificial ceremonies held for Mongolian emperors from the 13th century. It mainly consists of two major parts: the sacrificial ceremony of the Eight White Houses with the Great Khan's Royal Tent at the core and the sacrificial ceremony of Suld (military flag and emblem) (Figure 1). The sacrificial ceremony of the Eight White Houses has such forms as daily worship (once in the morning and once in the evening each day), monthly worship (on the first and third days of each lunar month), seasonal worship (once every four seasons of the year, on 21 March, 15 May, 12 September (Figure 2), and 3 October, respectively), and annual worship (on the first day of the first lunar month). In addition, there are multiple offerings to Genghis Khan, the Great Khan of the Mongols. The sacrificial ceremony of Suld also has such forms as daily, monthly, seasonal, and annual worship as well as fierce dragon year offerings (held every third day of the tenth month of the dragon year, once every twelve years) [71]. The sacrificial worship in the spring is the grandest and largest ceremony of the year, and the local government will hold public sacrificial ceremonies (Figure 3).

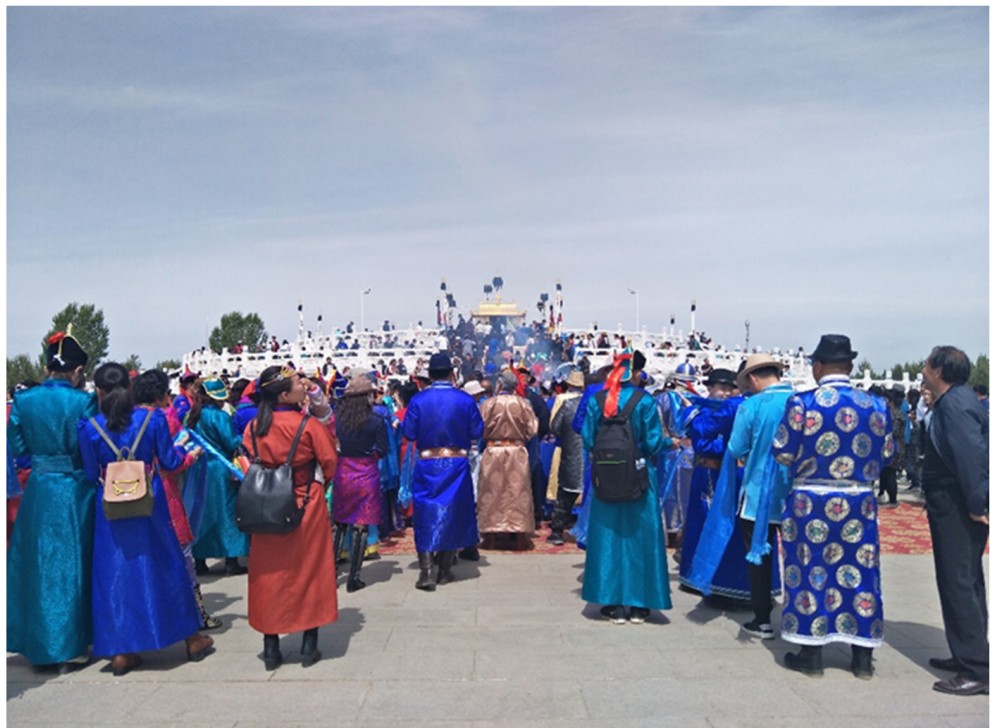

**Figure 1.** The Site of Suld sacrifice.

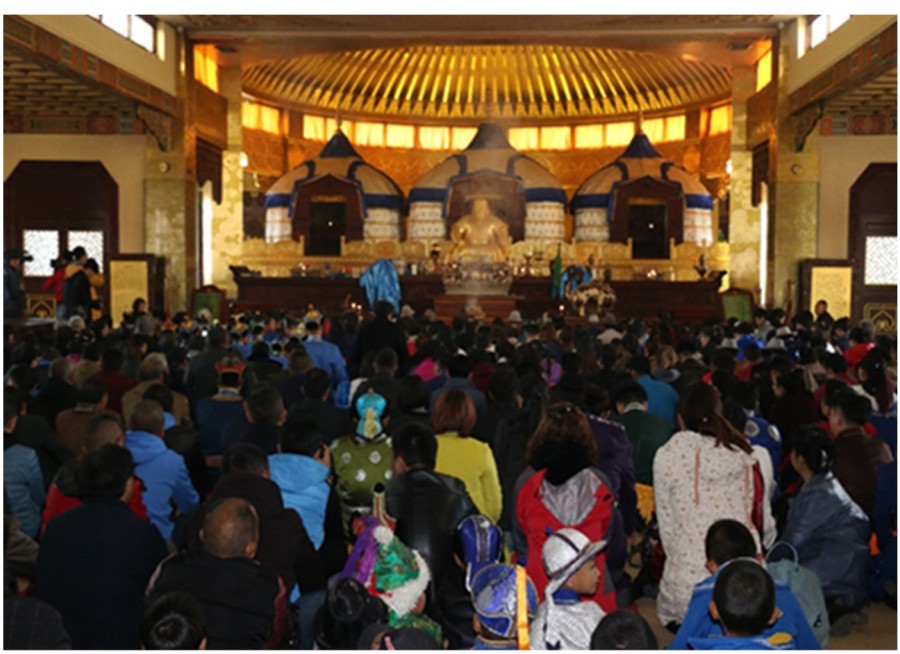

**Figure 2.** The Site of the Autumn Sacrificial Ceremony of Genghis Khan.

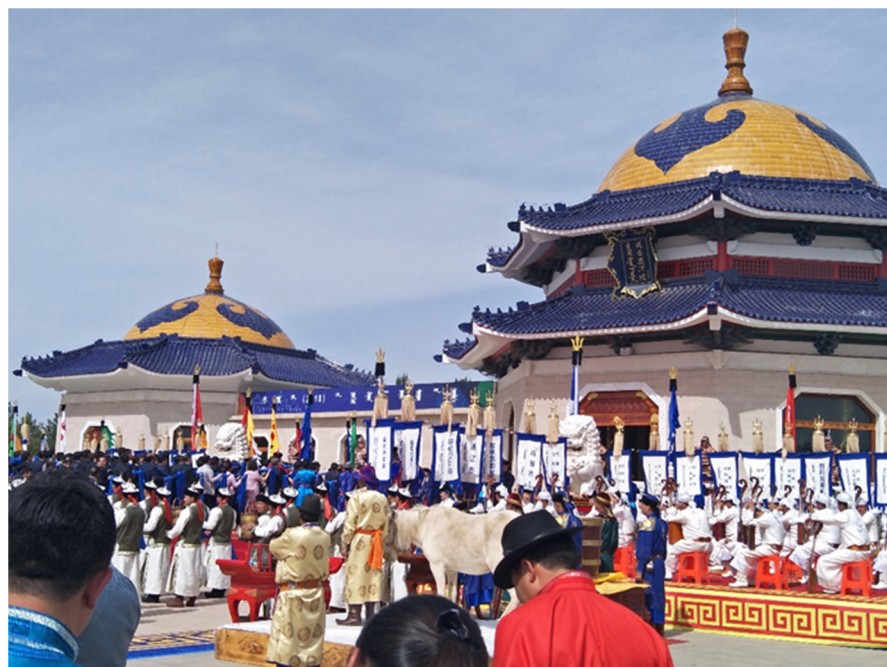

**Figure 3.** The Site of the Spring Sacrificial Ceremony of Genghis Khan.

Before the establishment of the People's Republic of China, the sacrificial ceremony of Genghis Khan continued in a movable form on the vast grasslands around its political center. After the establishment of the People's Republic of China, upon the approval of the central government in 1954, the Mausoleum of Genghis Khan was built, marking the relocation of the Eight White Houses to a singular, permanent place. From then on, the movable tents that lasted for more than 700 years were fixed. Located on the grassland of Ejin Horo Town, Ejin Horo Banner, Ordos City, Inner Mongolia Autonomous Region, China, the Mausoleum of Genghis Khan covers an area of approximately 5.5 hectares and consists of three Mongolian-style halls and connected corridors (Figure 4). The main building is magnificent and has a strong Mongolian ethnic style. It is divided into six parts:

the main hall, the bedroom hall, the east hall, the west hall, the east corridor, and the west corridor. The Mausoleum of Genghis Khan is of great value to the study of the history and culture of the Mongolian ethnic group and even the nomadic ethnic groups in northern China. It has been selected as a national key cultural relics protection unit and was included in the national list of the first batch of intangible cultural heritage announced by the State Council in 2006. The Mausoleum of Genghis Khan is viewed by the Mongols as the space for worship of Genghis Khan and is also the only place to show the sacrificial culture of Genghis Khan. Since 1985 when the Mausoleum of Genghis Khan was opened to the public, it has shifted from a secret sacrificial ceremony in which only people of the Mongol ethnic group can participate into an open sacrificial ceremony, indicating not only worshippers but also tourists can access the Mausoleum of Genghis Khan. With the accelerated development of the tourism industry supported by local governments, the sacrificial ceremony of Genghis Khan has also been commodified for tourism products, forming a tourism destination and having become a well-known tourism brand of Ordos City, with the core culture being that of paying tribute to Genghis Khan.

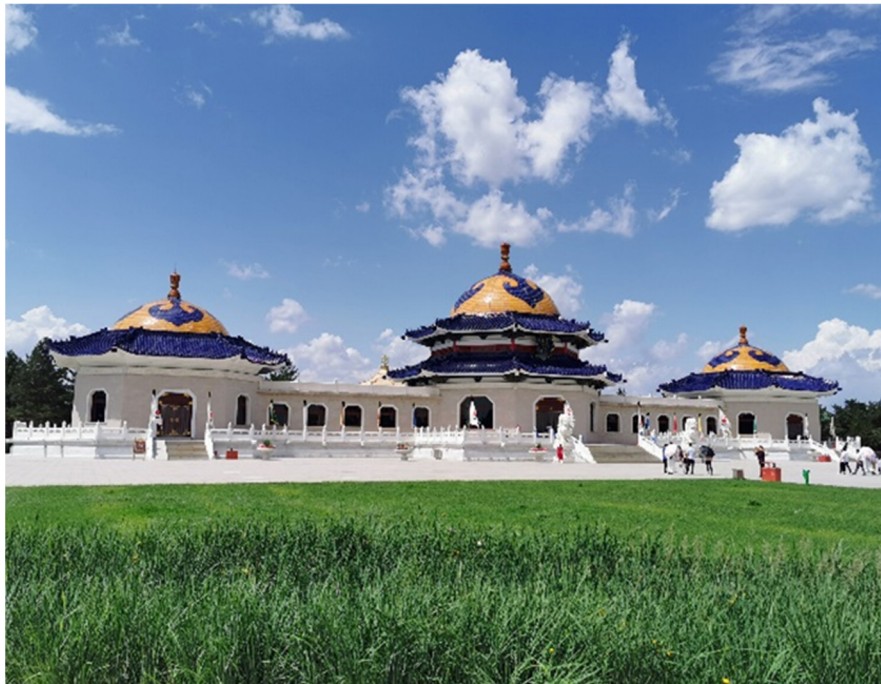

**Figure 4.** The Mausoleum of Genghis Khan.

### 3.2. Data Collection and Analysis

Through a literature review on tourism research, qualitative research methods were found to be the most commonly used methods to explore the topic of cultural commodification. Thus, in order to obtain a detailed and empirical understanding of the cultural experience in the process of tourism development, this study adopted a qualitative research method. The field investigation method was finally selected to understand and analyze cultural commodification for the account of an established concept; that is, a qualitative research design is more suitable for testing the complexity of tourism phenomena than a quantitative research design [72]. Moreover, analysis based on qualitative data can better test and understand phenomena [73].

The primary data in this study mainly came from field investigation, and the secondary data came from previous literature. The interviews were all conducted in Mongolian. The author is a Mongol and can speak the same language and has the same cultural background as the interviewees, thus can accurately capture the information provided by the interviewees and has the ability to translate interview materials into the required language, which also lays a good foundation for the smooth progress of this study. The

author conducted field investigations from October to November 2017 (29 days), May 2018 (27 days), July to August 2019 (14 days), and February 2023 (6 days). The first field investigation (2017) mainly focused on understanding the history of the sacrificial ceremony of Genghis Khan. The author further understood the culture by combining literature knowledge and communicating face-to-face with the Darkhad to understand their current situation and their understanding of the sacrificial ceremony of Genghis Khan. The second field investigation (2018) put emphasis on learning about the current status of tourism development from multiple sources and the author's personal attendance at the grandest sacrificial ceremony in the spring, during which the author observed the complete process of the sacrificial ceremony for a total of 5 h. The third field investigation (2019) was mainly intended to supplement the deficiencies or omissions in the previous two investigations. The fourth field investigation (2023) mainly aimed to learn about the sacrificial activities during the COVID-19 pandemic and the plans for sacrificial activities after the pandemic. The field investigations lasted 76 days, during which about 60 people were interviewed, including retired government cadres, scenic spot managers, dedicated guardians of the soul of Genghis Khan (Darkhad), personnel in charge of sacrificial affairs (professionals practicing sacrificial activities in the Mausoleum, belonging to the elites of the Darkhad), and other relevant personnel. The interviews covered the development history of the sacrificial culture of Genghis Khan, the current problems, the tourism development processes, and the current status of tourism development in the local area of the Mausoleum.

The field investigations were mainly conducted through face-to-face, in-depth interviews, and the snowball sampling method was used to select the respondents. The snowball sampling technique is a purposeful sampling form used when researchers have limited knowledge of the selected respondents [74]. In order to capture the complete interview, voice recording is required for most of the interviews provided that the interviewees grant permission, with some having refused to be recorded [75]. Therefore, the conversations with the respondents were recorded in this study after obtaining their consent. After the interview, the conversations were translated and transcribed into the corresponding language, with all interview texts manually encoded and divided into emerging themes and sub-themes.

Images can be used to explore the past in backward mapping [76]. In this study, visualization methods were also employed to strengthen the efficiency of traditional data collection methods. As a cultural checklist, images can provide a record of events and locations [77], allowing the author to consider and explore details in a careful and unhurried way.

## 4. Findings

After further exploration, it was found that Rousseau's framework would be of great benefit to the study of cultural influence. For the sake of gaining a more comprehensive understanding of tourism commodification, the author, based on the cultural layers model proposed by Rousseau [64] and adapted by Deery [67], decided to analyze the commodification levels of traditional culture from the perspectives of the following five levels: artifacts, patterns of behavior, institutional norms, values, and fundamental assumptions. The definitions of each level are as follows:

Artifacts mainly refer to various tourism handicrafts, souvenirs, etc., associated with the development of tourism. This is a relatively obvious manifestation of cultural commodification. The economic value of souvenirs rarely comes from their production costs but instead mainly comes from their "symbolic value" [78]. The commoditized culture is manifested as souvenirs or an experience for tourists with fetishism or even sacred attributes. Patterns of behavior mainly refer to the physical expression of traditional culture. In tourism research, it refers to the cultural expression displayed by the host to tourists with their gesture language, such as showcasing an exquisite craftsmanship to tourists. For a more accurate understanding and differentiation, it may be more appropriate to change the behavioral norms defined by Rousseau in organizational culture to institutional norms

for the account of the context of traditional culture in this study [64]. Institutional norms refer to the rules, norms, and taboos of traditional culture, which are the principles that culture holders collectively abide by in default, such as the specifications on the time, place, and frequency of rituals. Values refer to the mindset or orientation of culture holders in making judgments and distinguishing right from wrong, such as their tendency to judge or decide whether to use their culture for tourism development. Fundamental assumptions are cultural assumptions essential to the development of a certain culture, without which the development of the culture would not have been possible. For instance, the firm belief of the people of some tribes or communities that a local mountain is sacred and can bless their peace can be the fundamental assumption of a certain culture or the origin of that culture. The adapted cultural layers model employed in this study is shown in Figure 5.

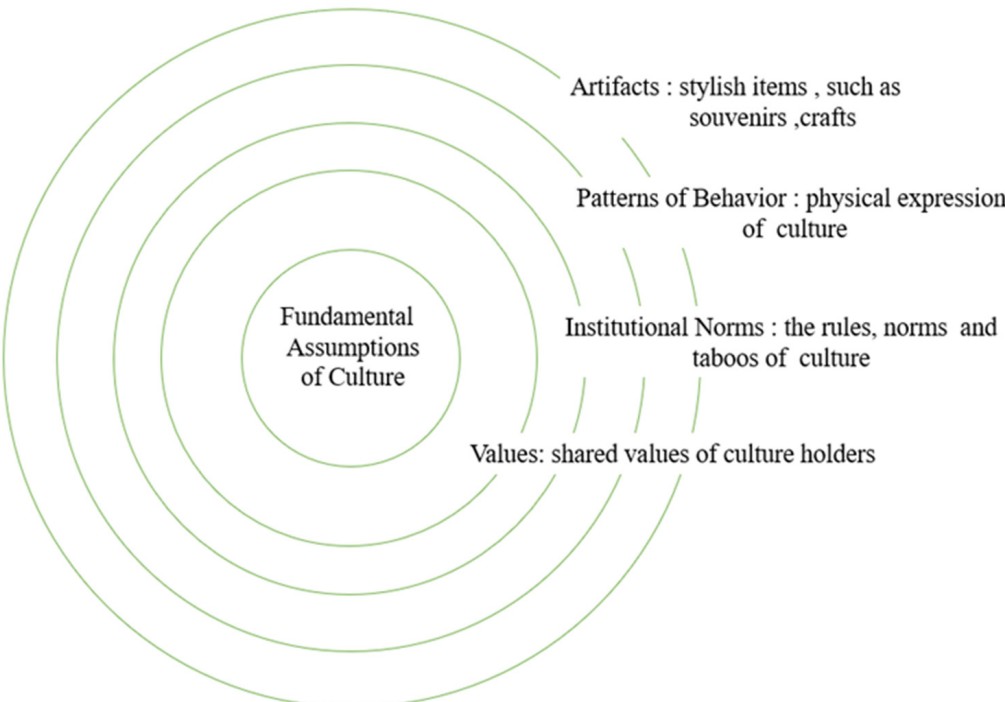

**Figure 5.** Layers of perceptions of cultural commodification. **Note:** Adapted from Rousseau [64] and Deery [67].

Based on the above-adapted model, the commodification level of the sacrificial culture of Genghis Khan will be analyzed in detail below.

*4.1. Commodification of Artifacts*

In the development process of the sacrificial culture of Genghis Khan, many precious sacrificial vessels with rich ethnic characteristics were produced. Some parts of sacrificial vessels were specially made by the guardians according to their sacrificial needs, and the other parts were worshipped by successive nobles and pilgrims. The sacrificial vessels, mainly made of gold, silver, and copper, are traditional sacrificial utensils of the Mongolian ethnic group. According to statistics in 1910, there were then a total of 834 sacrificial vessels in the Eight White Houses, including the Changming Lamp Cup, Milk Dispenser, Gold Wine Cup, Silver Lock, Silver Spoon, Large Wine Cup, Silver Bowl, Silver Plate, Silver Cup, Silver Wine Pot, Silver Basin, Fire Brace, etc. In the 17th century, when all the Mongol tribes paid allegiance to the Qing Dynasty, the Qing government adopted the policy of "the revival of shamanism to shape Mongolia" to consolidate and strengthen its rule over the Mongolian region, which contributed to the unprecedented flourishing of Tibetan Buddhism in the middle of the Qing Dynasty and the sharp increase of Tibetan Buddhism temples in Inner Mongolia [79]. The introduction of Buddhism culture to the

Mongolian region also influenced the sacrificial vessels used for the sacrificial ceremonies of Genghis Khan to some extent, and some temple sacrificial vessels were also used to worship Genghis Khan. For example, the sacrificial vessels still in use for the sacrifices to Genghis Khan include Dung (spiral horn), Zulcuuc (divine lamp cup), Humah (pure water bottle), holy water bottle, holy water cup, gong, fork, incense burner, etc. [70]. All the specialized ritual utensils need to be custom-made by local craftsmen; they are not available for sale in the market, and it is not allowed for the ritual utensils to enter the market as regular commercial products.

With the Mausoleum of Genghis Khan opening to the public and the rapid development of the tourism industry, the number of visiting tourists has shown a year-on-year increase. Visitors to destinations different from their own cultural traditions tend to have a strong will to bring tangible symbols representative of their experience back home. Tourists visiting the Mausoleum of Genghis Khan are not only interested in the local ancient and mysterious sacrificial culture but are also eager to purchase related souvenirs. A souvenir is a significant form of cultural commodification. In order to meet market demand, some souvenirs related to sacrifices to Genghis Khan and some sacrificial items have gradually emerged in the market of tourism souvenirs. Among the relevant tourist souvenirs, the most popular ones now are portraits of Genghis Khan (Figure 6). As for the main component of the sacrificial ceremony of Genghis Khan, the "Suld" (divine spear, flag, and banner) is a symbol of Genghis Khan's military power, and its replicas are popular tourist souvenirs (Figure 7). Among the specialized sacrificial vessels, only the Changming Lamp Cup (Figure 7) with textures such as copper, silver, and gold plating, as well as such common sacrificial or religious products as joss sticks, joss stick burners, incense burners, holy water cups, silver or copper wine cups and kettles) are currently on sale. Most of these tourist souvenirs are mass-produced in large factories outside the research area, while a small portion of them are locally crafted with distinctive local features by local artisans.

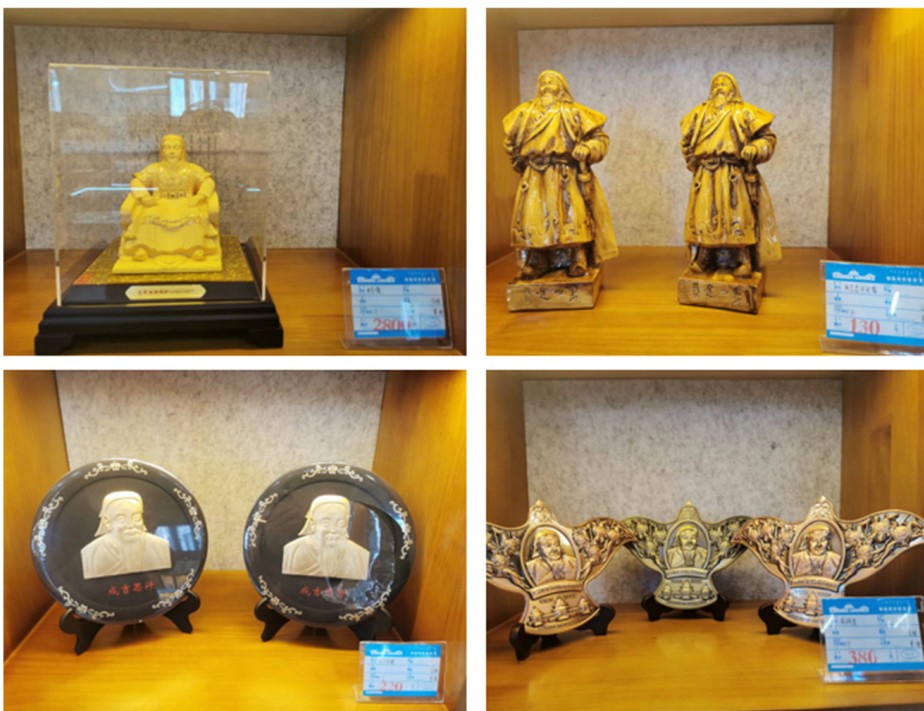

**Figure 6.** Some of the Genghis Khan portrait souvenirs.

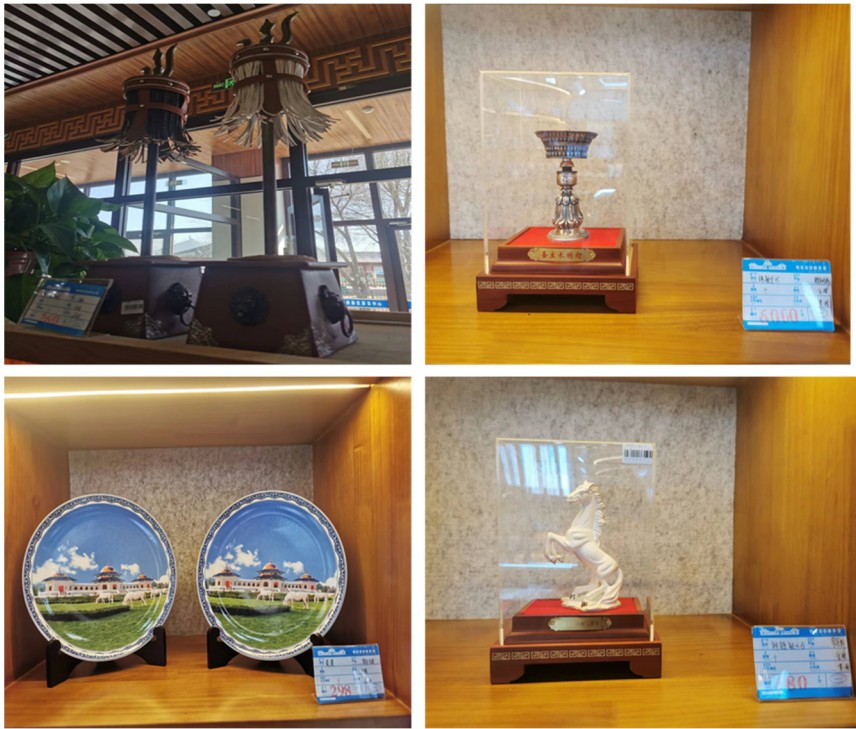

**Figure 7.** Some of the souvenirs and ritual items sold at the Genghis Khan Mausoleum.

Most buyers are pilgrims or admirers of Genghis Khan. The portrait of Genghis Khan may be hung at homes to pay respect to Genghis Khan or be given to family members and friends as a great gift. Changming Lamp Cups, joss stick burners, and incense burners are generally used to worship ancestors or gods at home. Some silver utensils are usually used as collections.

*Many tourists or worshipers visit this place and before leaving, they usually buy some souvenirs. The most popular items are figurines of Genghis Khan (Figure 6), as well as various souvenirs featuring Mongolian elements like horses. People who purchase larger portraits or items such as lanterns and Suld which are related to rituals, tend to buy fewer small souvenirs. Typically, these buyers are worshipers who take these items home to worship Genghis Khan.*

(Said by a Gift Shop Attendant at the tourist site)

In the commodification process of the sacrificial ceremony of Genghis Khan, there have not yet been any items that contradict the cultural connotations at the level of artifacts.

*4.2. Commodification of Patterns of Behavior*

The sacrifices to Genghis Khan may consist of numerous sacrificial ceremonies throughout the year, each consisting of a series of rituals, such as the Hada offering ceremony, the Holy Lamp offering ceremony, the whole sheep offering ceremony, the Holy Wine offering ceremony, the Sacrificial Song offering ceremony, the Sacred Flame offering ceremony, the Blessing ceremony, the Blessing-bringing ceremony, the Benison ceremony, the Blessing song ceremony, etc. Each ritual has its own specialized function and significance. Some rituals are open to the public and can be watched by tourists, while others are not publicly available and cannot be attended or watched by visitors. Therefore, the sacrificial ceremony of Genghis Khan allowed for tourism development and is also a part in which visitors or worshippers can participate.

In most cases, tourists may not be content with just purchasing souvenirs and may have a strong interest in visiting or experiencing the unique traditional practices and ceremonies of the local culture. During the visit and experience of the sacrificial ceremony

of Genghis Khan, tourists can observe or partially participate in the daily, monthly, seasonal, and annual sacrificial activities, as well as personally practice sacrificial offerings. Sacrificial offering is a ritual performed by pilgrims at any time to express their sincere wish for Genghis Khan. The detailed process usually involves the following: the Darkhad on duty leads the worshippers to worship Genghis Khan by consecutively offering Hada, a holy lamp, and holy wine. The worshiper kneels down with his own offerings in his/her hands, facing the Great Khan's Royal Tent. After receiving the offerings, the presiding guardian drapes Hada over the worshiper, passes on the name of the worshiper to the Great Khan (for allowing him to know who the worshiper is), recites blessing words for the worshiper, and then serves the holy wine offered to the Great Khan for the worshiper to drink. People believe that the wine offered to the Great Khan can ward off disasters, eliminate diseases, and bring peace. The worshiper kowtows three times to the Great Khan's Royal Tent and then leaves the site. People who come to worship generally offer butter, brick tea, dairy products, white wine, Uuch (the notum of the ram), or an entire ram to Genghis Khan, the Great Khan of the Mongols, or make donations. Mongolian people believe that butter, brick tea, etc. are superior food, while Uuch (the notum of the ram) is the noblest food and is commonly used by Mongols for sacrificial ceremonies and entertaining guests. The common practice is to put boiled whole sheep on a special plate, with the four legs on the bottom side and the sheep's back and head on the other side.

With the growing number of visitors following the development of tourism, the income from sacrificial ceremonies, which can be viewed as tourism income to some extent, is increasing year by year, having now become an important income source of the Genghis Khan Mausoleum Tourist Area. Some of the worshippers are devout pilgrims, while others are seeking paid tourism experiences, which are also the main forms of commodification of sacrifices to Genghis Khan. As a priest said: "*With the development of tourism, the worshippers are not limited to our own ethnic groups. We have noted and welcome other ethnic groups and even foreigners coming to worship the Great Khan devoutly. No matter who comes to worship, we will complete all the steps according to traditional rituals. We have no prescriptions for offerings and donations. People may make offerings and give donations in whatever way they want to*".

> *Meanwhile, there is an evident phenomenon that as the number of worshipers increases, the workload of the priest also increases. Perhaps in order to provide more blessings (experience) to worshipers or tourists, the priest have shortened their prayers compared to before the development of tourism, while still maintaining their authenticity. This can also be seen as a change in behavior patterns caused by the tourism commodification.*

> *They have discretionary power over the donations made by the visitors, and there is no specific amount for those contributions. They can at least retain a portion of the money internally, so they actively engage with the tourists as it adds to their income. Previously, the prayers used to last for more than ten minutes, but now, in order to accommodate more tourists, they are completed in just three or four minutes . . .*

> (said by culture holders)

### 4.3. Adhere to Institutional Norms

Although the development of tourism brings an unprecedented opportunity to monetize the sacrifices to Genghis Khan, the provisions on traditional time, scale, form, and taboo of sacrifices to Genghis Khan are consistently adhered to under the Darkhad's guard. They do not cater to the tourism market to develop tourism products in the form of staging performances and adapting the content and form of the rituals.

> *A few (scenic areas) leaders mentioned the idea of giving priority to performance before. We all set ourselves against performance because it is not a part of our culture. If we give way to performances, we will deceive our Great Khan and tourists at the same time. We stipulate the time of the sacrifice and need to strictly implement it according to the rules. Some leaders pay attention to culture and others lay emphasis on the development*

*of tourism. It is impossible to gain from performances when there are only tourists and no worshipers.*

(said by a priest)

*We oppose immoderate staged performances and dramatization and must comply with regulations for sacrificial ceremony. Surrounded by this idea, it is allowed to add some programs for the sacrifice ceremony as long as they do not clash with the content of the sacrificial ceremony. In the past, we opposed some stage shows which were mostly naked in the scenic area because we are utterly opposed to the content which clashes with the sacrifice taboos.*

(said by culture holders)

There are many taboos in the sacrificial ceremony, like the Garli sacrifice in which the Altan Urugh's descendants of Genghis Khan make sacrifices in the form of burning food for ancestral spirits on the night before the grand ceremony in spring (on the evening of lunar 20 March) in specialized place. The guardians of the youngest son, Tolui of Genghis Khan, host the Garli sacrifice without the participation of others. *Dedicated persons are responsible for some sacrifices, so it is not advised to participate in these sacrifices (even the author has not taken part in Garli sacrifice in the over 20 years' life there). These sacrifices are for dedicated persons and we can't participate and break their taboo. Although as a leader, the author hasn't been in Garli sacrifice for all these years (said by a retired government leader).*

In addition, the Darkhad persistently insists on the uniqueness of inheritance for sacrifices to Genghis Khan. The Darkhad is a guardian group shouldering a sacred mission that is specially organized and responsible for guarding, worshipping, managing, and migrating royal tents during the religious sacrifices to Genghis Khan that were established after his death. They have been carrying out sacrifices to Genghis Khan for nearly eight hundred years through the hereditary family system and became a special and unique group directly inheriting duties related to the sacrificial ceremonies of Genghis Khan among the Mongolian ethnic group, keeping the mysterious sacrifice intact in the Ordos until now. The Darkhad is divided into Darkhad to Great Khan and Darkhad to Suld according to the responsibility they undertake. Darkhad to Great Khan (the descendants of one of Genghis Khan's nine senior generals, Boersch) is mainly in charge of and carries out sacrificial activities for the Great Khan. Darkhad to Suld (the descendants of one of Khan's nine senior generals, Muqali) is mainly in charge of and carries out sacrificial activities for Suld. The Darkhad has had a strict organizational system since its foundation, and both sides of the Darkhad are, respectively, separated into eight functional departments, each of which is equipped with three groups. Each department and group have a clear division of labor and bear different duties and functions in the process of the ceremony. The male offspring of the clade is entitled to inheritance by heredity from generation to generation. For example, in the sacrificial ceremony, people who are responsible for instrumental music playing, cooking the whole sheep, burning joss sticks, placing offerings, serving wine, or guarding the door all have a clear division of labor. People who occupy the job are chosen through inheritance and undertake this job from generation to generation.

*We have a clear division of labor in this sacrifice. Since ancient times, the family who is in charge of the job will do this job from generation to generation, which cannot be replaced by each other and only inherited by own offspring. If there is no male offspring in the family, or if they cannot fulfill their obligations for some special reasons, someone else from the clan will be selected after consultation and going through a series of formalities. Fathers teach sons concrete operations from generation to generation.*

(said by a priest)

It can be seen that the traditional institutional structure of sacrifices to Genghis Khan has not changed with the development of tourism. So far, there has not been a staging performance of a sacrifice ceremony for tourists to gain more experience or increase tourism income. Culture holders still adhere to the original system of traditional sacrifice. Only

when culture holders can master their own destiny can cultural change lead to a new form of authenticity [80].

*4.4. The Values of the Darkhad: Loyalty and Protection*

The sacrifices to Genghis Khan have been passed on for nearly 800 years so far. It is true that the inheritance of the sacrificial culture of Genghis Khan by the descendants of Genghis Khan, despite 800 years of difficulties and obstacles, is indispensable to the support and respect of all dynasties and the government of the People's Republic of China, but the persistent protection of the Mongolian ethnic group, especially the Darkhad, is the key. Culture holders are the masters of culture and deciders of cultural fate to some extent. Whether culture can be preserved and inherited depends on external environmental conditions on the one hand and the wills of culture holders on the other hand.

It is not a smooth or easy process for the Darkhad to stick to the sacrifices of Genghis Khan. During the Great Mongol Nation Period and the Yuan Dynasty, sacrifices to Genghis Khan were fully supported and developed; in the Northern Yuan Dynasty, the descendants of the Yuan Dynasty retreated to Mobei and established a new regime that was against the Ming Dynasty. When the army of the Ming Dynasty approached all the way northward, the Darkhad strongly continued to make sacrifices to Genghis Khan in adversity by virtue of Erdo (royal tent)'s removable advantage. In the Qing Dynasty, the sacrifices to Genghis Khan were deprived of the status of national sacrifice and became a folk custom. Under the jurisdiction of the Court of Colonial Affairs of the Qing Dynasty, when Zinon (the highest officer of the Darkhad)'s power was reduced, and the government lacked money, five hundred households of Darkhad (only five hundred households were allowed in the Qing Dynasty) raised five hundred taels of silver to prepare offerings and cover the expenses through an annual contribution of a tael of silver per household. The Darkhad collected alms all over Mongolia to address problems of expenses they needed when updating or repairing their facility when necessary. During the period of the Republic of China, the Eight White Houses were under the jurisdiction of the Kuomintang government. The government's decision to move the Eight White Houses was agreed upon due to the war and the invasion of Japan, so the Eight White Houses were moved to Xinglong Mountain in Yuzhong County, Gansu Province, in 1939 with the Kuomintang escort and moved to Qinghai Tar Temple on the eve of the liberation of Lanzhou in 1949 [76]. During the westward migration, the Darkhad asked to go with them and changed shifts once a year. Due to the long journey and lack of transportation, it was very common to encounter robbers on the way back to Ordos when walking for more than 40 days [69]. After the founding of the People's Republic of China, with the approval of the State Council, the Mausoleum of Genghis Khan was built in 1954 and moved from Qinghai Province and back to the ancestral site Ordos in 1956. Since then, the state of unfixed sacrifice has been ended, and people have begun to make concentrated sacrifices, thus achieving a safe and stable development environment.

But what has been making the Darkhad firmly stick to it for nearly 800 years? The answer is their values of being loyal to the Great Khan. The core values of the Darkhad can be summarized as loyalty to Mongke Tangri and the Great Khan, which stems from Mongolian beliefs. Mongols worship Mongke Tangri and Genghis Khan. They equate both and believe that God and the Great Khan are aware and it is their responsibility given by God to protect the spirit of Genghis Khan. If they do not sacrifice to Great Khan, they will be punished by God. Therefore, the Darkhad are loyal to the Great Khan and fulfill the task assigned by God and dare not slack off from generation to generation.

> *This is our task. We have been doing this job on behalf of all Mongols for eight hundred years. Will heaven and earth forgive us if we stop doing this? Every generation ponders over this problem because this is a task given by God and the sacrifice to Genghis Khan is related to God.*

(said by a priest)

*We think that the spirit of Great Khan gives us this sacred task and the mission shows us the bright way. When we were young, our elders in the family told us that our greatest task is to sacrifice to Great Khan and protect the spirit of Great Khan. We cannot slack off. Perhaps it is this kind of deep-rooted education that passed on this sacrificial culture from generation to generation.*

(said by culture holders)

*Mongols believe that God is aware. If you do bad things, maybe others do not know, but God is visible, God will punish you and Great Khan is also watching. Therefore, when we do something and then pray to Great Khan, we will think that Great Khan must bless us to achieve our dreams.*

(said by culture holders)

*The biggest belief of Mongols is to sacrifice to heaven. To sacrifice to Genghis Khan is to sacrifice to heaven. Genghis Khan represents God having perception, so Genghis Khan also has perception. Therefore, we dare not disobey, or Great Khan and God will punish them. Therefore, we are respectful to Great Khan and God, and also fearful of them.*

(said by a priest)

They always adhere to such values. Although considerable economic benefits are brought by tourism development, their belief cannot be shaken. Therefore, there has been no change in values to cater to the needs of tourism development until now.

*4.5. Fundamental Assumptions: The Spirit of the Great Khan Is Perceptive*

The animistic view of the ancient nomads in the north is the cultural root of the formation of the sacrifices to Genghis Khan, which is the integration of Shamanism, Tibetan Buddhism, Central Plains Han culture, and other diverse cultures. The ancient Mongols, who believed in Shamanism, regarded Mongke Tangri as the supreme God and Genghis Khan as the favored one of God. This is the reason why mysterious sacrifices to Genghis Khan arose after the death of Genghis Khan. Shamanism holds that the soul is immortal after death. The Mongols believe that Genghis Khan's divine soul must live on in another form after his death. Therefore, Genghis Khan's last breath was absorbed in white camel wool and preserved for worship.

The fundamental assumption of the sacrificial ceremony of Genghis Khan is embodied in two aspects. On the one hand, it has important modeling or symbolic significance and is a dramatic expression of religious thoughts. On the other hand, it is the means of communication with spiritual objects and the ability of the professionals practicing sacrificial activities to summon gods, attach them to the "offerings", obtain divine edicts from them, and then convey people's wishes to gods. In the sacrifices to Genghis Khan, the professionals practicing sacrificial activities act as living media between man and God. They offer the sacrificial tributes or transfer the sacrificial tributes of worshipers.

Through a series of prayers, the sacrifice is attached to spirituality and accompanied by corresponding language and action to complete the sacrificial ceremony. Also, they communicate with gods by virtue of the professionals practicing sacrificial activities. Our worship ceremony is different from Buddhism. It requires an intermediary between humans and gods; without this intermediary, the worship cannot be completed (said by culture holders).

In addition, Mongols respected the sun and fire in nature exceedingly, which is closely related to their grassland environment and lifestyle. Fire can keep out the cold and resist the attack of wild animals. In the sacrificial ceremony, fire is the medium of spiritual communication between man and God, and Mongols also hope to promote the rebirth of the soul.

Therefore, the fundamental assumption of the sacrificial culture of Genghis Khan is that later generations regard Genghis Khan as the favored one of God so as to express their respect for ancestors and heroes. Under the influence of ancient Shamanism, people believe that their souls will not perish after death. It is believed that Genghis Khan's soul will bless

later generations in another form, and they can communicate with the Great Khan through the medium between man and God and obtain the grace of the Great Khan.

*The worship of Genghis Khan has been going on for nearly 800 years, and it is not an ordinary matter. Firstly, it has received support from various dynasties. Why is that? Because it is a task bestowed by the heavens, and the worship of Genghis Khan is connected to the divine. Nowadays, materialism is advocated, and it is said that there is no existence of the soul. However, if one lacks faith and does not believe in the existence of the soul, then they are merely walking corpses. The actions of the Darhut people are about elevating the soul. If there is no belief and recognition of the existence of the soul, then the work we have done for hundreds of years would be in vain, as if we have done nothing at all.*

(said by a priest)

With the constantly deepening influence of modernization and, especially, the rapid development of tourism, many traditional ideas have been replaced by so-called more scientific new ideas. However, the guardians of the sacrifices to Genghis Khan always adhered to the original values and beliefs, revered the Great Khan, and were not swept up by this wave of modern thought.

## 5. Discussion

Taking sacrifices to Genghis Khan as an example, this paper discusses layered structures of tourism commodification. The existing literature discusses tourism commodification from the perspective of all or part of the elements. In view of the diversity of the contents contained in culture, this study emphasizes the view that the discussion of cultural commodification should be divided into levels and examines the degree of the commodification of sacrifices to Genghis Khan through the five levels of the cultural layers theory. A wider applicability of the model has also been attempted.

### 5.1. Degree of Commodification in Genghis Khan Sacrifice

Given the previous research on cultural commodification is insufficient, the cultural layers theory of organizational culture research is introduced into the study of cultural commodification. At present, the commodification level of the sacrifices to Genghis Khan consists of artifacts and patterns of behavior. Since the sacrifices to Genghis Khan contributed to the development of tourism, culture holders always have adhered to the original time and institutional norms of the sacrifice and made no changes to cater to the development need of tourism, although both the government and tourists hope to see regular stage performances. Culture holders also adhere to the values of loyalty to the Great Khan and fulfilling the divine mission, and always believe in the fundamental assumption that the soul is immortal after death and that the Great Khan is perceptive in heaven.

### 5.2. The Broader Applicability of Cultural Layers Theory in Tourism Commodification Research

The cultural layers theory introduced in this study is not only applicable to the analysis of the commodification degree of the sacrifices to Genghis Khan but may be extended to more cases in the study of cultural commodification. This study tried to stratify the degree of commodification according to the cases in the previous literature regarding cultural commodification, and the results are as follows (Table 1). Then, this study tried to explain the bases of some cases. In the case of "Traditional basket weaving in Botswana" [23], the study examined the impact of tourism development on traditional basket weaving in the Okavango Delta, Botswana. The article described a gradual shift in knitting—from a domestic and practical product to one of the most sought-after products in the tourist market. The study concluded that although changes are inevitable, local communities are urged to strike a balance between meeting the needs of the market and preserving the cultural significance of knitting. The baskets have changed from articles of daily use to tourist artifacts. At the same time, the shapes and sizes of artifacts are changed to meet

the needs of tourists, but deeper cultural issues are not involved, so they are considered instances of the commodification of artifacts. In the case of "The Songkran Festival in Thailand" [81], the study explored the transformation of cultural practices through the interplay between a hallmark cultural event, tourism, and commercial activities. Although the case has changed from a traditional cultural activity to an activity with commercial value, its original system and norms have not changed and just are open for tourists to participate in, according to the above text. Therefore, it can be considered that its commodification level belongs to patterns of behavior. In the case of "The Land Diving Ritual in Vanuatu" [19], the study revealed how tourism heightens community tensions when traditional culture is commercialized. It is also mentioned in the article that this case was originally a ceremony of special significance conducted in the ethnic group. After commodification, its original holding time was extended, and many secrets were also disclosed. Therefore, it might be reasonable to consider its commodification level as being on the level of institutional norms. In the case of "Goo-Moremi Gorge in Botswana" [26], the study analyzed the process of tourism commodification of Goo-Moremi Gorge, which is considered sacred by Goo-Moremi Village, as well as the emergence of ambivalence, tension, and local resistance. Some locals accept the commodification of their culture as tourism products for the sake of the expected socio-economic benefits. In this case, some people in the local community no longer consider the valley sacrosanct for reaping economic interests and commoditizing the sacred mountain into tourist products. Therefore, the degree of commodification belongs to the level of values. The last analysis case was an early classic case, which set off the boom of cultural commodification research. "The Alarde of Fuenterrabia in Spain" [44] is a case written sorrowfully by the famous anthropologist Greenwood after long tracking. The ceremony has been one of the hot spots of tourist activities in Spain. With the development of tourism, the local government regards this grand ceremony as a public event and arranges frequent performances to attract tourists. At this point, the exciting celebration ritual was gone. This case illustrates that the symbol of solidarity turned into a purely commercial activity, then into a political activity, and finally into a tragedy because of the changes in fundamental assumptions of the culture. It also shows that if the fundamental assumptions of culture, the roots of culture, are commodified, culture will eventually disappear. The above is a tentative analysis of the levels of commodification in different cases, which aims at illustrating the wide applicability of the cultural layers theory.

**Table 1.** A tentative analysis of other cases.

| No. | Case | Layer of Artifacts | Layer of Patterns of Behavior | Layer of Institutional Norms | Layer of Values | Layer of Fundamental Assumptions | Related Research |
|---|---|---|---|---|---|---|---|
| 1 | Maya culture in Belize | | P | | | | Medina [7] |
| 2 | Balinese culture of Indonesia | P | | | | | Pitana [9] |
| 3 | Zapotec Indian textile of Mexico | P | | | | | Cohen [12] |
| 4 | Gulag performances in Kazakhstani | | | P | | | Tiberghien & Lennon [41] |
| 5 | Goo-Moremi Gorge in Botswana | | | | P | | Mbaiwa [26] |
| 6 | Adat Ceremony of Eastern Indonesia | | P | | | | Cole [17] |
| 7 | Shaolin Monastery of China | | | | P | | Hung [25] |
| 8 | The Land Diving Ritual in Vanuatu | | | P | | | Cheer [19] |
| 9 | Traditional basket weaving in Botswana | P | | | | | Mochankana et al. [23] |

**Table 1.** *Cont.*

| No. | Case | Layer of Artifacts | Layer of Patterns of Behavior | Layer of Institutional Norms | Layer of Values | Layer of Fundamental Assumptions | Related Research |
|---|---|---|---|---|---|---|---|
| 10 | The Alarde of Fuenterrabia in Spain | | | | | P | Greenwood [43] |
| 11 | Dong traditional music in China | | P | | | | Song & Yuan [49] |
| 12 | The Yi ethnic group's "Duze" (Torch Festival) in China | | | | P | | Li & Peng [82] |
| 13 | The Songkran Festival in Thailand | | P | | | | Intason et al. [81] |
| 14 | Toraja's funeral tradition in Indonesia | | P | | | | Crystal [83] |
| 15 | The Dai ethnic group's Water Splashing Festival at Xishuangbanna, China | | | P | | | Wu & Zhu, 2015 [84] Wu & Yuan [85] |

*5.3. Misalignment between the Sequence of Tourism Commodification and Cultural Layer Sequence*

The significance of the stratification of cultural commodification lies in revealing the priority of various elements in culture rather than the order of cultural commodification. Cultural commodification does not necessarily occur from the outside in. For example, the behavioral pattern layer may be commoditized first, followed by related souvenirs. The commodification of some cultures may even belong to the commodification of the institutional level at the very beginning, which changes the original time, place, and frequency of the culture to meet the need of the tourism market for the sake of tourism development. However, it is worth noting that the deeper level of commodification inevitably leads to the superficial level of commodification. For example, if the commodification level of a culture belongs to the institutional level, then the artifacts and patterns of behavior layers must have been commoditized. At the same time, the influence of tourism commodification deepens with the deepening of commodification level, and different levels of commodification reflect different social problems. Therefore, the logical coordination of cultural elements themselves plays a role in the process of cultural commodification. When we use cultural layers for analysis, we should pay attention to the mutual influence between cultural elements rather than one-way change.

In a word, tourism commodification is a complicated problem. Different cases have different backgrounds, processes, forms, logic, and implied deep meanings of commodification. Therefore, using layered thinking to view the case can not only analyze the degree of commodification in a more comprehensive and clear way but also reveal the relationship and function between elements in the process of commodification, providing a basis for guiding tourism practice more effectively, as well as a reference for the study of cultural sustainability.

To sum up, the main contribution of this study is to introduce the cultural layers theory to analyze tourism commodification, which provides a new research approach in this field. The cultural layers theory can solve the debate on the advantages and disadvantages of cultural commodification caused by the analysis of culture as a whole in previous studies. For example, tourism commodification may save a traditional skill on the verge of extinction and revive traditional culture. Thus, according to the study results of such cases, some researchers believe that the influence of cultural commodification is positive. However, other researchers see that tourism commodification changes the values and even fundamental assumptions of traditional culture, thus leading to the acculturation or disappearance of traditional culture. In such instances, they think that the influence of tourism commodification is negative. This is because they do not treat culture in a layered way since the levels of commodification in these two cases are fundamentally different. Therefore, the analysis of tourism commodification with layered thinking can not only

reveal the levels of tourism commodification but also solve the debate on the pros and cons of tourism commodification in previous studies. It is also a reference for resolving various social contradictions caused by commodification so as to better guide tourism practice.

## 6. Conclusions

Firstly, this research sheds light on cultural sustainability through the analysis of tourism commodification. Through the introduction of cultural layers theory, this paper analyzes the layers of tourism commodification in sacrifices to Genghis Khan and reveals its characteristics of commodification. At present, commodification mainly occurs in the layers of artifacts and patterns of behavior, which are still in the shallow layers, whereas the deeper layers of institutional structures, values, and fundamental assumptions have not yet been commoditized.

Secondly, the cultural layers theory can not only reveal the layers of tourism commodification but also more comprehensively reflect the relationship and interaction between the cultural elements. The process of tourism commodification does not necessarily follow the sequence of cultural division from the outside to the inside; instead, the deeper layer of commodification will inevitably drive the commodification of its outer circle, and different layers of commodification display different social problems as well.

Finally, it is found that this theory is also suitable for other tourism commodification cases unfolded in existing studies. Therefore, the theory is also a way to solve the debate over the pros and cons of tourism commodification and provide the basis for guiding the regulation of tourism commodification more effectively. The focus on different layers of local culture provides a more comprehensive understanding of commodification and the protection of cultural heritages and thus offers insight into cultural sustainability in practice.

## 7. Limitations and Further Studies

It is worth mentioning that this study, while stratifying culture and structuring it in tourism commodification studies, can reveal a clearer and more comprehensive picture of the extent of tourism commodification, and the case used in this study is well suited to a stratified discussion. However, layered thinking also inevitably fragments the connections between layers within the culture to some extent; after all, sometimes, the boundaries between cultural layers are not necessarily clear. In addition, the stratification judgment of other cases is only based on relevant literature without field investigation. Therefore, the layer attributed may not be accurate. The main purpose of Table 1 is to expand the applicability of the cultural layers theory for the study of cultural commodification and to provide a more intuitive perception. The analysis and summary of the complex process and logic of cultural commodification need longer tracking and a wider range of cases. This is the motivation and direction for further research.

**Author Contributions:** Conceptualization, investigation, and writing—original draft, L.B.; validation, writing—review, editing, and funding acquisition, S.W. All authors have read and agreed to the published version of the manuscript.

**Funding:** This work was supported by the National Natural Science Foundation of China [41201139].

**Informed Consent Statement:** Informed consent was obtained from all subjects involved in the study.

**Data Availability Statement:** Not applicable.

**Conflicts of Interest:** The authors declare no conflict of interest.

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
