# Peer review of "New Perspective of Cultural Sustainability: Exploring Tourism Commodification and Cultural Layers"

_sustainability, doi:10.3390/su15139880_

Round 1

Reviewer 1 Report

I would like to say thanks to the authors for the good research and good presentation of its results. The authors presented a very interesting study for the general public. The article is written according to all the requirements for a manuscript for publication in rating journals; it is written so well that you can clearly see the logic of the study; the methodology is described in detail; the conclusions are logical and are based on the conducted research.

However, in my opinion, the biggest drawback of the article is that it belongs to a journal. After all, the word Sustainability is found only in the title and abstract. Further throughout the text, Sustainability is never mentioned.

Nevertheless, I would like to suggest several ways to improve the manuscript:

Point 1: A literature review is sufficient; however the article could be added about the image of the region, for example. You can use this one https://doi.org/10.32933/ActaInnovations.43.6

Point 2Mention the article’s limitations

Point 3: it is worth adding a discussion section a comparison of their research with those already existing in this area and additionally show the novelty of the research and what contribution it makes to the theoretical/methodological and/or practical aspects.

Author Response

The response to the reviewers has also been uploaded as an attachment to the PDF.

Manuscript: sustainability-2420869        

Responses to Reviewer’ Comments

Dear reviewer,

We would like to thank you for constructive comments on our article. These comments are all valuable and helpful for improving our article. All the authors have seriously discussed about all these comments. According to the comments, we have tried best to modify our manuscript to meet with the requirements of the journal. In this revised version, changes to our manuscript within the document were all using track changes mode. Point-by-point responses to the reviewers are listed below.

Point 1: A literature review is sufficient; however, the article could be added about the image of the region, for example. You can use this one https://doi.org/10.32933/ActaInnovations.43.6

Response: Thank you for recommending the important literature. We have carefully read the mentioned paper, and its main points are indeed crucial for enhancing our article. We have cited it in the literature review, specifically referring to line 166-168.

Point 2Mention the article’s limitations

Response: Thank you for your kind comment. Previously, discussing the limitations of our study within the discussion section might have led to a less clear structure and insufficient emphasis on the related aspects. We have added a Section 7 in the article, specifically addressing the limitations and future research directions. It can be found in lines 790 to 802. By doing so, we aim to improve the clarity of the article's structure and give more prominence to the relevant content. In this study, while stratifying culture and structuring it in tourism commodification studies can reveal a clearer and more comprehensive picture of the extent of tourism commodification, and the case used in this study are well suited to a stratified discussion. However, a layered thinking also inevitably fragments the connections between layers within culture to some extent; after all, sometimes the boundaries between cultural layers are not necessarily clear. In addition, the stratification judgment of other cases is only based on relevant literature without field investigation. Therefore, the layer attributed to may not be accurate. The analysis and summary of the complex process and logic of cultural commodification need longer tracking and a wider range of cases. This is the motivation and direction for further research.

Point 3: it is worth adding a discussion section a comparison of their research with those already existing in this area and additionally show the novelty of the research and what contribution it makes to the theoretical/methodological and/or practical aspects.

Response: Thank you for your kind comment. The discussion section of the article became somewhat lengthy, which may have resulted in a lack of clarity in the hierarchical structure, making it difficult for readers to quickly grasp some important points. Following your suggestion, we have now supplemented and improved the presentation of the novelty of our study and its contributions to theory and practice. The specific additions can be found in lines 753 to 768 of the article. The main contribution of this study is to introduce the cultural layers theory to analyze the tourism commodification, which provides a new research approach in this field. The cultural layers theory can solve the debate on the advantages and disadvantages of cultural commodification caused by the analysis of culture as a whole in previous studies. The analysis of tourism commodification with layered thinking can not only reveal the levels of tourism commodification, but also solve the debate on pros and cons of tourism commodification in previous studies. It is also a reference for resolving various social contradictions caused by commodification, so as to better guide tourism practice.

 Point 4:It is mentioned in the comment that the word Sustainability is found only in the title and abstract, further throughout the text, Sustainability is never mentioned.

Response: Thank you for your kind comment. This is a very important issue, as the entire article should be closely aligned with the title and the focus of the journal. Moreover, our research aims to provide a novel research perspective for cultural sustainability in the context of tourism. We overlooked this aspect in our initial submission. In this revision, we have added content related to cultural sustainability in the introduction, discussion, and conclusion sections to align with the theme. Specifically, the additions can be found at L22, L27-31, L84-87, L751-752, L770-771, and L786-789 of the article.

We are very appreciating all your comments and favorable consideration. Thank you for helping us to identify the problems in the article. We would be honored and delighted to learn from you. I hope that these revisions and the improved text will be satisfactory and make the paper be acceptable for publication in “sustainability”.

Kind regards,                                                        

Sincerely yours,

Lingxiao Bai

Shixiu Weng

Reviewer 2 Report

The reviewed article presents a very broad description of an extremely interesting local tradition and describes the theory of tourism commodification and shows the development of local culture.

In my opinion, the article contains too broad a description of local customs and too few results from interviews with respondents.

Whether commodification is shown in the article? Tourists only attend events as spectators, they participate in ceremonies only as viewers.

There is no explanation in the description of whether the souvenirs prepared for these events are prepared by local craftsmen in accordance with all the rules, or whether they are mass-produced in factories that are located outside the study area.

The article also lacks an explanation of what the entire tourist facilities look like, whether it is specially prepared so that as many accommodate as possible can come in, or whether existing local facilities are used. What does the preparation of this place for tourists look like, tourist development, tourist facilities, etc.

The author should focus on the Discussion, because it is a description of similar cases, this part should be included in the section describing similar cases, in the theoretical part. This should be a discussion, not a case study.

I believe that the author should rethink the structure of the presented study or the content described.

Author Response

The response to the reviewers has also been uploaded as an attachment to the PDF.

Manuscript: sustainability-2420869        

Responses to Reviewer’ Comments

Dear reviewer,

We would like to thank you for constructive comments on our article. These comments are all valuable and helpful for improving our article. All the authors have seriously discussed about all these comments. According to the comments, we have tried best to modify our manuscript to meet with the requirements of the journal. In this revised version, changes to our manuscript within the document were all by using track changes mode. Point-by-point responses to the reviewers are listed below.

1.In my opinion, the article contains too broad a description of local customs and too few results from interviews with respondents.

Response: Thank you for your kind comment. The field investigation for this study spanned from 2017 to 2023, covering a period of 7 years and involving over 60 interviews. However, in this paper, due to the focus of the main theme and considering the limitations of the article's length, only a representative portion of the interviewees' responses is presented. Considering the proportion of their inclusion in the text, the information from the interviews may indeed appear somewhat limited. Based on your suggestion, additional interview content has been incorporated into the article, enhancing the supporting materials and making the arguments more persuasive. The corresponding additions can be found in L413-418, L479-483, L630-632, and L644-651.

2.Whether commodification is shown in the article? Tourists only attend events as spectators, they participate in ceremonies only as viewers.

Response: Thank you for your kind comment. The tourism commodification of Genghis Khan sacrifice primarily manifests in the commodification of artifacts and behavioral patterns. The commodification of artifacts mainly involves the production and sale of Genghis Khan-related souvenirs and ritual items within the tourism context. The commodification of behavioral patterns is mainly reflected in the worship of Genghis Khan. Since the ceremonial rituals associated with Genghis Khan are not commercially performed, if visitors want to experience the worship process, the only way is to bring their own sacrificial offerings or donate alms to participate in the worship experience. This is also the main channel for priests and the scenic area to increase their income through commodification. Therefore, at present, the phenomenon of commodification is quite evident. Specific details have been supplemented and improved in sections 4.1 and 4.2 of the original text. Please refer to L473-478 for the specific additions.

In addition, as the ritual ceremonies have not been developed into tourism products, regular visitors cannot witness large-scale ceremonies during their normal visits. The process of the ritual ceremonies can only be observed on designated days of worship. Therefore, for visitors who wish to have a more immersive experience beyond observing the ritual ceremonies, they can participate in the experience of the priests’ offering prayers to Genghis Khan by making sacrificial offerings or donating alms. As a result, visitors to the Mausoleum of Genghis Khan can witness the ritual ceremonies (if they visit on designated worship days) and can also obtain the experience of worshipping Genghis Khan through offering sacrifices. Detailed information about this can be found in section 4.2.

3.There is no explanation in the description of whether the souvenirs prepared for these events are prepared by local craftsmen in accordance with all the rules, or whether they are mass-produced in factories that are located outside the study area.

Response: Thank you for your kind comment. The ceremonial tools exclusively used for Genghis Khan's sacrifice are custom-made by local craftsmen and are not available for sale on the market. They are also not allowed to enter the market as regular commodities. The majority of tourist souvenirs are mass-produced in large factories outside the research area, while a small portion of them are locally crafted to showcase ethnic characteristics. Your question will indeed be a common concern among readers. Therefore, relevant explanations have been added to the article, which can be found in L385-388 and L404-407. Additionally, in order to visually present the related tourist souvenirs, Figures 6 and 7 have been included in the text.

4.The article also lacks an explanation of what the entire tourist facilities look like, whether it is specially prepared so that as many accommodate as possible can come in, or whether existing local facilities are used. What does the preparation of this place for tourists look like, tourist development, tourist facilities, etc.

Response: Thank you for your kind comment. The Mausoleum of Genghis Khan was initially established in 1956, before which it followed the nomadic lifestyle of the Mongolian people, moving across the grasslands. After its construction, the mausoleum became permanently located in present-day Ordos. With the flourishing development of the tourism industry, the local government, aiming to promote tourism, expanded the original mausoleum into the current tourist attraction in the early 2000s. The expansion primarily involved the addition of sculptures and murals depicting the life and battle scenes during the era of Genghis Khan, along with the construction of necessary facilities for the tourist site. The mausoleum, completed in 1956, remains unchanged in its structure. However, a tourist area has been constructed around it, accompanied by the addition of tourism facilities. Large-scale ritual ceremonies are conducted outdoors, while smaller and daily rituals take place inside the mausoleum, which can accommodate 200-300 people. I apologize for the lack of additional images in the article to visually showcase the scenes of the tourist area, as the concern was to keep the length of the article manageable.

5.The author should focus on the Discussion, because it is a description of similar cases, this part should be included in the section describing similar cases, in the theoretical part. This should be a discussion, not a case study.

Response: Thank you for your kind comment. The purpose of this paper is to analysis the case of the Genghis Khan sacrifice in an attempt to show that the introduction of cultural layered thinking in the study of tourism commodification can reveal a fuller and clearer picture of the extent of tourism commodification.We have also provided corresponding discussions in the article. Firstly, the framework chosen for this study helps us better understand the different levels of commodification. In the discussion, we have also utilized selected cases from previous research on tourism commodification (Table 1 in the article) to analyze their levels of commodification. We found that the theory of cultural layers not only applies to analyzing the case of Genghis Khan sacrifice in this study but also has broader applicability to analyzing other cases. Therefore, we believe that this framework has wider relevance. Secondly, the significance of categorizing cultural commodification into different layers is to reveal the prioritization of various elements within culture, rather than establishing a sequential order of cultural commodification. Cultural commodification does not necessarily occur in a linear progression from outer to inner layers. In some cases, the commodification process may begin at a deeper level even before its integration into tourism, such as instances where traditional customs are altered to cater to tourists. Thirdly, categorizing cultural commodification into different levels allows for a clearer assessment of the degree of commodification. Previous studies often analyzed culture as a whole, which made it challenging to evaluate or compare the level of cultural commodification. However, adopting a hierarchical perspective in examining case studies enables a more comprehensive and distinct analysis of the degree of commodification. It also reveals the relationships and interactions among different elements during the commodification process, providing a basis for effectively guiding tourism practices and offering insights for research on cultural sustainability.

       Perhaps our discussion section was a bit lengthy, making it difficult for readers to quickly grasp the main points of each viewpoint. Following your suggestion, subheadings have been added to the "5. Discussion" section in the article to enhance clarity and readability.

6. I believe that the author should rethink the structure of the presented study or the content described.

Response: Thank you for your kind comment. The structure of this study is arranged according to the logic of empirical research. The core sections are organized based on the framework's progression from surface to depth, effectively showcasing the role of the framework in the article. Additionally, through the analysis of five levels and in-depth interpretation of case studies, the degree of commodification is presented. Therefore, we have adopted this framework in the writing of our paper. While this framework may not be perfect, it does, to some extent, present the material and analytical logic in response to the research questions stated in the introduction. We greatly appreciate the point raised by the expert, and we have indeed carefully reconsidered the framework issue. However, we have not yet found a better solution to address it, so we are currently continuing to use the original framework.

We are very appreciating all your comments and favorable consideration. Thank you for identify the problems in the article. We would be honored and delighted to learn from you. I hope that these revisions and the improved text will be satisfactory and make the paper be acceptable for publication in “Sustainability”.

Kind regards,                                                        

Sincerely yours,

Lingxiao Bai

Shixiu Weng

Round 2

Reviewer 1 Report

The authors made a great job. I suggest to publish the research.